# Optimization of irrigation scheduling for maize in arid regions Northwest China based on water stress diagnosis in models

Lina Zhou[1,2], Jianxin Jin[1,2]*, Jinyin Lei[1,2]

1 Agricultural Resources and Environment Research Institute, Ningxia Academy of Agriculture and Forestry Sciences, Yinchuan, China, 2 Yinchuan Observation Station of National Agricultural Environment, Yinchuan, China

* jinnxnk009@163.com

## Abstract

Water stress is an important factor affecting maize yield in arid areas. By optimizing the irrigation scheduling for maize based on water stress conditions, precise and efficient irrigation could be achieved. Based on the sensitivity analysis, calibration, and validation of parameters in the Soil Water Air Plant (SWAP) model, a SWAP-IES assimilation simulation system was constructed by integrating SWAP and IES (Iterative ensemble smoother). The water stress coefficient Ws was calculated to reflect the water stress status experienced by maize throughout its entire growth period, and the irrigation amount and cycle of the experimental design were adjusted according to Ws. During the calibration and validation process, the SWAP model achieved a high level of accuracy in simulating maize LAI, plant height, biomass, and yield, with NRMSE ranging from 8.01% to 18.37%. The experiment showed that the Ws of maize in each treatment had remained below 1 throughout the entire growth period, indicating that the maize had been in a state of water stress, especially after 80 days of emergence, with Ws ranging from 0.07–0.60. According to different strategies, the optimized maize irrigation quota in the Yellow River irrigation area of Ningxia was 357.29 mm − 462.55 mm, and the irrigation water utilization efficiency was 2.82 kg/m$^3$ - 3.37 kg/m$^3$, achieving the goal of water conservation and high efficiency.

Irrigation determined the sustainable development of agricultural production in arid and semi-arid regions and was an important means of stabilizing and increasing grain production [1]. In China, maize served as the predominant grain crop. It had consistently sustained a stable sowing area exceeding 40 million hectares, a situation that holds profound significance in safeguarding the self-sufficiency of vital agricultural products and ensuring national food security [2]. In recent years, extreme weather frequency increased, large rainfall spatiotemporal shifts occurred, and intense long – lasting droughts led to water stress, which severely impacts maize production,

**Data availability statement:** The original data used in this paper is uploaded in the Supporting information files.

**Funding:** Ningxia Agricultural Science and Technology Innovation Guidance Project (NKYJ-25-03).

**Competing interests:** The authors have declared that no competing interests exist.

causing stunted growth, assimilation capacity reduction, and yield decrease [3]. For a long time, people had been able to resist the adverse impacts of climate conditions through irrigation, achieved high yields, and ensured stable food production.

A reasonable irrigation scheduling was crucial for ensuring water supply during the critical water demand period of maize and reducing the impact of water stress on yield. Many scholars had carried out a lot of work on optimizing the irrigation scheduling for maize, proposing irrigation scheduling optimization methods based on field experiments, water balance principles, crop growth models, etc., and forming irrigation strategies for different regional climates, soils, cultivation modes, etc [4]. The traditional field experiment method selected suitable irrigation schedules by setting irrigation gradients or water deficit at different growth stages, and comparing them with the crop growth and yield under sufficient irrigation [5]. It was also possible to evaluate the differences in maize growth and irrigation water use efficiency under different irrigation methods such as surface irrigation, drip irrigation, and ditch irrigation, and the conclusions obtained were more intuitive and reliable [6]. Soil moisture was the direct source of water for maize. Based on the relationship between soil moisture and the water demand pattern of maize, the water balance of farmland and the deficit of maize root water absorption were quantified. The optimized irrigation scheduling could significantly improve the yield and irrigation efficiency of maize [7]. Irrigation management could also be achieved by integrating crop and soil indicators, such as considering crop water stress index and relative soil moisture content with certain weights, and setting different irrigation thresholds at different growth stages of maize to obtain efficient and water-saving irrigation schedules [8].

The traditional field experiment method, due to defects in experimental design and field operation, not only required a large amount of experimental work, but also could not obtain accurate irrigation amount and time. The crop growth model could be based on the principles of crop physiology, ecology, and environmental science, and quantitatively describe the process of crop growth and development through mathematical methods. It had a wide range of applications in the optimization of maize irrigation schedules [9]. By using crop growth models to optimize irrigation, multiple indicators such as soil water storage, evapotranspiration (ET), and soil moisture content could be selected as evaluation indicators. By setting different lower limits for irrigation indicators at each growth stage, an optimized irrigation scheduling could be proposed, which could achieve multi-objective improvements such as yield and water use efficiency [10–13]. Crop growth models could also be used to simulate and optimize suitable irrigation strategies for different crop varieties under changes in carbon dioxide concentration and temperature, in order to evaluate the adaptability of varieties and irrigation to climate change [14].

Due to crop variety characteristics, soil spatial variability, and simplified assumptions of the model, it was easy to encounter situations where the calculated irrigation amount was insufficient or excessive when using crop growth models to optimize irrigation schedules. Some scholars had proposed using data assimilation methods to assimilate field observation data into crop growth models to reduce simulation errors, synchronize the simulation of maize growth processes with irrigation decisions, and

improve the scientific and reliable irrigation strategies [15]. For example, Li et al. significantly improved the yield by using assimilation algorithms to integrate measured soil moisture into crop growth models and then combining these modified soil moisture simulation values with farmer experience to search for the optimal irrigation system under certain constraints [16]. Li et al. assimilated the LAI obtained from remote sensing into the CERES-Maize model and used a multi-objective genetic algorithm to optimize irrigation schedules under various meteorological scenarios, achieving the goals of water conservation, yield increase, and improving water use efficiency [15]. When optimizing irrigation schedules using crop growth models, in order to reduce the randomness of typical annual meteorological data, long time series meteorological data could be selected for simulation. Based on a large number of scenario simulations, singular values could be eliminated to calculate the arithmetic mean or other weighted methods could be used to obtain optimized irrigation schedules for different rainfall year types [17,18]. The crop growth model could also achieve quantitative simulation of water consumption processes such as evaporation, drainage, and runoff in farmland, and reduce water resource waste caused by ineffective water consumption processes through scenario simulation. It could optimize the irrigation amount and time throughout the entire growth period of maize, greatly improving the irrigation water use efficiency [19]. It was evident that crop growth models offered significant advantages in optimizing maize irrigation schedules. They could substantially reduce workload, enhance work efficiency, and allow for re-simulation and optimization based on different crop varieties, cultivation patterns, farmland hydrological conditions, and climate change.

However, existing studies primarily relied on indicators such as soil water storage, soil moisture, and evapotranspiration when utilizing crop growth assimilation simulation systems to optimize maize irrigation scheduling. Alternatively, these studies adjusted the irrigation time and amount throughout the entire growth period based on the target yield. This resulted in it being impossible to irrigate precisely according to crops' water-shortage information. Furthermore, the proposed irrigation scheduling lacked real-time dynamic adjustment capabilities in line with the meteorological conditions of that year, which led to an inability to accurately align the irrigation amount with the water requirements of maize. Therefore, this study reflected the water stress status of maize by simulating the water-stress coefficient (the ratio of actual transpiration to potential transpiration) and proposed an efficient method for formulating irrigation schedules. Not only did this approach provide a scientific basis for the precise and efficient management of maize water resources in arid and semi-arid regions, but it also offered reference methods for optimizing irrigation schedules for other crops.

## 1. Materials and methods

### 1.1 Experimental design

**(1) Overview of the experimental area.** The experiment was carried out in Wanghong Town, Yongning County, Ningxia, spanning 2019–2020. The experimental area is located at 106°11′E longitude and 38°10′N latitude, with an altitude of 1,125 meters. It features a typical continental semi-arid climate, with an average annual rainfall of 198.19 mm and an average annual evaporation of 1,186.73 mm [20]. The soil in the area is sandy loam, and the cultivated layer has an average bulk density of 1.45 g/cm$^3$. The available nitrogen, phosphorus, and potassium contents, as well as the organic matter content in the cultivated layer of the experimental site, are 85.77 mg/kg, 12.31 mg/kg, 125.84 mg/kg, and 2.3%, respectively. The meteorological parameters of the experimental area are shown in S1 Fig.

**(2) Experimental design.** The tested maize variety was Xianyu335, and this experiment was a single-factor experiment focusing on irrigation amount. Based on the irrigation practices of local farmers, a total of nine gradient irrigation amount treatments were set up, with irrigation quotas of 1800 m$^3$/ha (T1), 2160 m$^3$/ha (T2), 2520 m$^3$/ha (T3), 2880 m$^3$/ha (T4), 3240 m$^3$/ha (T5), 3600 m$^3$/ha (T6), 3960 m$^3$/ha (T7), 4320 m$^3$/ha (T8), and 4680 m$^3$/ha (T9). As shown in S1 Table, each treatment had an irrigation frequency of 8 times. The experimental plot was 10 meters long and 5 meters wide, covering an area of 50 square meters. There were three replicates for each plot, resulting in a total of 27 plots. The sowing distance between maize sown plants was 25 cm, and the row spacing was 60 cm. Drip irrigation was employed as the irrigation method. The drip irrigation tape within each plot was arranged in a single straight line, with a dripper spacing

of 0.3 m and a dripper flow rate of 2.0 L/h. A water meter was installed at the head of each plot to strictly control the irrigation amount. The total amount of fertilizer applied was 320 kg/ha of nitrogen (N), 120 kg/ha of phosphorus ($P_2O_5$), and 252 kg/ha of potassium ($K_2O$). This included a basal application of 30% nitrogen fertilizer and 70% phosphorus-potassium fertilizer. The remaining nutrients were applied in water during the large horn stage, tasseling stage, and filling stage, respectively. Other field operations remained consistent.

**(3) Measurement indicators and methods.**

① Soil moisture content: Before and after irrigation, the drying and weighing method was used to determine the soil moisture content. The 0–100 cm soil layer was divided into 10 layers at 10 cm intervals, and soil samples were taken using a soil drill. After weighing the fresh soil samples, they were placed in a 105°C oven to dry to a constant weight, and then were weighed again to calculate the soil moisture content. Additional testing has been done after rainfall, which is to determine the soil moisture content once according to the aforementioned method.

② Plant height and biomass: Plant height and biomass were to be measured once at each growth stage of maize, and a total of 5 measurements were taken throughout the entire growth period. Plant height was measured using a tape measure, three maize plants were randomly selected from each plot, and the arithmetic mean of the plant heights from the ground surface to the growth point was calculated. After measuring, the maize plants were dug out and brought back to the laboratory for washing. The oven was adjusted to 105°C and the plants were withered for 30 minutes, after which they were dried at 80°C until a constant weight was reached. Then they were weighed using an analytical balance.

③ Leaf Area Index (LAI): It was measured using the LAI-2200 canopy analyzer, during each growth period of maize, measurements were taken under backlight from 16:00–18:00 on clear weather. Three measurements were taken for each plot, and the arithmetic mean was taken.

④ Yield: During the physiological maturity stage, 3 sampling points were randomly selected from each plot, with a 10 row spacing for each sampling point. A representative 20 meter double row was selected from the 10 rows, and plants and ears were counted. The number of ears per hectare was calculated. An ear was collected every 5 ears in each measurement segment, and a total of 20 ears were harvested as samples to determine the number of grains per ear. After all the grains were removed, they were weighed, dried, and then weighed again to calculate the grain moisture content. The average value was taken for each replicate and it was converted into the yield (kg) per hectare.

### 1.2 Construction of assimilation simulation system

**(1) Model calibration method.** Soil Water Atmosphere Plant (SWAP) model is a mechanistic model developed by Wageningen University in the Netherlands. It was designed to simulate water movement, solute transport, heat transfer, and crop growth processes in soil plant atmosphere continuous systems (SPAC) [21]. According to the sensitivity analysis results of model parameters, field measurements and trial-and-error methods were used to calibrate parameters with strong sensitivity, while model default or reference methods were used to calibrate parameters with low sensitivity. Based on the observed data of LAI, biomass, soil moisture content, and yield from the 2019 irrigation experiment, the model parameters were adjusted and calibrated to achieve global optimization of simulation results for all treatments from T1 to T9. The specific verification method was described in the following text [22]. The established parameter set was validated using the experimental results from 2020.

① accumulated temperature from emergence to flowering (TSUMEA) and Accumulated temperature from flowering to maturity (TSUMAM): they were calculated using temperature data observed at the meteorological station of the experimental base and the biological lower limit temperature of maize.

② initial biomass (TDWI): Sampling was conducted in the field during the two leaf and one heart stage of maize, and the aboveground parts were cut with scissors, dried, and weighed before being converted into initial biomass per hectare based on seedling emergence rate.

③ LAI at emergence (LAIEM): Samples were taken in the field during the two leaf and one heart stage of maize, and all leaves were cut off. The length and width were measured using a ruler, and the leaf area and LAIEM were calculated using the length width coefficient method (full leaf coefficient of 0.75, incomplete leaf coefficient of 0.5).

④ Specific Leaf Area (SLA): During the maize seedling stage, large horn stage, and grain-filling stage, three fully unfolded leaves were taken from each maize plant from top to bottom, and the leaf area was determined using the length-width coefficient method. After the leaves were blanched in an 80 ℃ oven for 30 min, the oven temperature was adjusted to 105 ℃ and the leaves were dried to a constant weight. Then they were weighed with an electronic balance to calculate the SLA of maize in different stages.

⑤ the methods for obtaining parameters of other major crops are shown in S1 Appendix.

**(2) Parameter sensitivity analysis.** The Morris sensitivity analysis method was used to analyze the sensitivity of input parameters in the SWAP model. This method was a global sensitivity analysis method proposed by Morris in 1991 [23]. After continuous improvement and refinement, it had significant advantages in the sensitivity analysis of models with multiple parameters, complex structures, and large computational workloads. It could achieve a good balance between computational efficiency and accuracy. The calculation results of this method could simultaneously provide the sensitivity of individual parameters and the interaction between parameters with respect to the output results of the model. Assuming the model had n parameters, namely $x_1, x_2, x_3 ...... x_n$, the model calculation result could be expressed as $y = f(x) = f(x_1, x_2, x_3 ...... x_n)$. First, the parameters of the model were normalized. The values of each parameter were projected onto the matrix of [0, 1], and the matrix of [0, 1] was discretized into p levels with Δ as the step size. In this way, the model parameters formed an $n*p$ spatial array, which constituted the sampling space for Monte Carlo random values of the model parameters [24]. In a simulation process, it was assumed that the randomly selected parameter samples were $x_1^1, x_2^1 ...... x_n^j$, then the parameter sensitivity index was as follows:

$$\mu_i^* = \frac{\sum_{j=1}^{m} \left| [y(x_1^1, x_2^1 ... x_i^1 + \Delta ... x_n^j) - y(x_1^1, x_2^1 ... x_i^1 ... x_n^j)]/\Delta \right|}{m} \tag{1}$$

In the formula, $\mu_i^*$ represented the parameter sensitivity index when the *i*-th parameter was disturbed. The larger its value was, the more sensitive the parameter was. *J* was the parameter set formed after the *j*-th resampling of the n × p sampling space. *I* was the parameter number corresponding to the disturbance Δ that occurred after the *j*-th resampling. m was the number of resampling times during the sensitivity analysis process. The crop parameters of the SWAP model varied greatly among different crop types and varieties, and might even change under different field management and production conditions. Soil was affected by field micro-topography, soil stratification, human activities, and other factors. The values of soil physical properties, hydraulic characteristics, and other parameters often exhibited variability at the field scale. Moreover, crop parameters and soil parameters were important parameters that affected the accuracy of model simulation. Therefore, this sensitivity analysis mainly analyzed two types of parameters: crop parameters and soil parameters. By referring to the literature, the range of values for the main parameters of maize was obtained, and an uncertainty of ±10% above and below the average value was set as the random sampling spatial range. The Morris global sensitivity analysis method was used for qualitative analysis. A total of 59 SWAP model input parameters were selected for sensitivity analysis, including 54 crop parameters and 5 soil parameters. The quantity level p was set to 4, and the number of parameter samples was r = 5900. The sensitivity of leaf area index (LAI), plant height (PH), and yield to these parameters was tested separately.

**(3) SWAP-IES Assimilation Simulation System.** Iterative ensemble smoother (IES) algorithm was a continuous data assimilation method that simultaneously utilized all time-series data [25]. Its assimilation idea originated from the optimization method of posterior probability density function sampling. This method estimated the sensitivity information of parameter states using a sample set and iteratively updated parameters using the Gaussian Newton method. It was assumed that the relationship between the input and output of the model could be expressed as:

$$d_{obs} = F(m) + \varepsilon \tag{2}$$

In the formula, $d_{obs}$ was the measured value vector, F was the forward model, m was the unknown parameter vector, and $\varepsilon$ was the observation error vector that conformed to a mean of 0 and a covariance of $CD = E[\varepsilon\varepsilon^T]$. Then IES could be implemented in the following steps. Step 1: $N_e$ samples were generated from the prior distribution of parameters to form the initial sample set $M_0$.

$$M^0 = [m_1^0, ..., m_{N_e}^0] \tag{3}$$

The superscript in the formula represented the iteration order, and the subscript represented the sample number. Step 2: In the *l*-th iteration, where $l = 1,2,\cdots$. Given historical observation data at all times, the parameter sample $M_l$ could be updated according to the following equation:

$$M^{l+1} = \beta_l M^0 + (1-\beta_l)M^l - \beta_l C_M G_l^T (C_D + G_l C_M G_l^T)^{-1} \times [F(M^l) - d_{obs} - G_l(M^l - M^0)] \tag{4}$$

In the formula, $\beta_l$ was the parameter for adjusting the iteration step size, $CM = \triangle M_0(\triangle M_0)^T/(N_e-1)$ represented the prior covariance of the parameter, which remained unchanged throughout the entire iteration process. $\Delta M_0$ represented the deviation between matrix $M_0$ and its mean. $G_l$ was a sensitivity matrix based on set-averaging, which played a role in connecting the input and output changes of the model. Step 3: Step 2 was repeated until the convergence standard or the iteration count set by IES was reached.

The flowchart of the constructed SWAP-IES assimilation yield estimation system as shown in S2 Fig. Using LAI and SW as observation variables of the SWAP-IES, a total of 6 observations were made throughout the entire growth stage. It was assumed that the soil was homogeneous in the horizontal direction, and all crop parameters, except for uncertain parameters, remained unchanged under different water stress treatments. Based on the sensitivity analysis results, five parameters, namely TSUMEA, CVO, SPAN, EFF, and CVL, were selected as the uncertainty parameters to be corrected and dynamically adjusted during the assimilation simulation. During the assimilation process, each parameter generated a prior value set with variances of 0.04, 0.08, 0.1, 0.24, and 0.08, respectively, and the ensemble size was set to 100. The observation variables used for data assimilation, such as LAI and SW, were measured data from field water stress experiments. Gaussian noise was used to perturb the state variables with variances of 0.1 and 0.01, respectively, to generate a sample set of observation values. The assimilation gain was calculated using the sample set, and the uncertainty parameters in the model were adjusted repeatedly until the preset convergence criteria were met.

## 1.3 Optimization strategy for irrigation scheduling

In order to achieve precise irrigation, the water stress coefficient Ws was used to control the amount and time of irrigation. The irrigation principle was that no water stress or only mild water stress occurred in each growth stage of maize. Ws was set at two lower limits of 0.9 and 0.95, and the irrigation trigger threshold for each growth stage was set according to the water demand pattern of maize. The irrigation amount was determined by irrigating to a non-stress state (i.e., Ws = 1). In

order to facilitate field management and avoid frequent irrigation operations, three irrigation decision gradients were set up for simulation calculation, as shown in S2 Table.

## 1.4 Evaluation index

The accuracy and consistency of the model simulation were tested using three indicators: the coefficient of determination ($R^2$), the root mean square error (RMSE), and the normalized root mean square error (NRMSE). The closer $R^2$ was to 1 and the smaller the RMSE, the higher the simulation accuracy was. NRMSE≤10% indicated extremely high accuracy; 10%<NRMSE≤20% indicated high accuracy; 20%<NRMSE≤30% indicated moderate accuracy; and NRMSE> 30% indicated low accuracy. The calculation formulas for each indicator were as follows:

$$R^2 = 1 - \frac{\sum_{i=1}^{n} (x_i - y_i)^2}{\sum_{i=1}^{n} (x_i - \overline{x_i})^2} \tag{5}$$

$$RMSE = \sqrt{\frac{\sum_{i=1}^{n} (x_i - y_i)^2}{n}} \tag{6}$$

$$NRMSE = \frac{\sqrt{\frac{\sum_{i=1}^{n} (x_i - y_i)^2}{n}}}{\overline{x_i}} \tag{7}$$

Among them, $x_i$ represented the measured value, $y_i$ was the simulated value, $\overline{x}$ was the mean of the measured values, and n was the sample size. The water stress coefficient ($W_s$) was used to represent the degree of water stress experienced by crops, and the calculation formula was as follows:

$$W_S = \frac{T_a}{T_p} \tag{8}$$

In the formula: $T_a$ was the actual daily evaporation of maize, mm; $T_p$ was the actual daily evaporation of maize, mm。 Using irrigation water utilization efficiency to evaluate the production capacity of irrigation water, the calculation method is as follows:

$$IWUE = \frac{Y_a}{I_a} \tag{9}$$

In the formula: $Y_a$ was the grain yield of maize, kg/ha; $I_a$ was the total irrigation amount of maize, m³/ha.

## 2. Results

### 2.1 Sensitivity analysis results

The sensitivity of maize LAI, plant height, and yield to the main parameters of the model were shown in S3 Fig. The soil parameters that were mainly sensitive to LAI included NPAR (shape coefficient of soil moisture characteristic curve) and OSAT (saturated soil moisture content), with sensitive indicators μ* of 0.16 and 0.10, respectively. Among crop parameters, the main parameters that were sensitive included TSUMEA, RGRLAI (maximum daily relative growth rate of leaf area index), SPAN (leaf life span at 35 °C), SLA, EFF (leaf light use efficiency), etc. The sensitivity index μ* of each parameter ranged from 0.19 to 0.82. The yield was mainly sensitive to crop parameters, which were SPAN, EFF, CVO (transfer efficiency of dry matter to storage organs), TSUMEA, FO (dry matter allocation ratio of storage organs), AMAX (maximum $CO_2$ assimilation

rate of leaves), etc., in order of sensitivity. The sensitive indicators μ* ranged from 0.31 to 1.51. Plant height was most sensitive to PHMAX (potential plant height), followed by PHA and PHB (parameters in the plant height growth curve), with sensitive indicators μ* of 18.93, 2.16, and 2.05, respectively. NPAR and OSAT in soil parameters were the most sensitive, with sensitive indicators μ* of 7.28 and 2.96, respectively. It could be seen that soil parameters, which had a significant impact on soil moisture availability and retention capacity, were sensitive to simulation results. Similarly, crop parameters that significantly affected maize nutritional growth days, potential leaf survival cycles, leaf growth rates, photosynthetic assimilation ability, assimilate accumulation rates, assimilate distribution ratios, and conversion efficiency were also sensitive to simulation results. Therefore, during the application of the model, it was necessary to focus on verifying these sensitive parameters.

## 2.2 Calibration and validation of SWAP model

### 2.2.1 Simulation of growth indicators.
The SWAP model was calibrated and validated using the results of maize irrigation experiments in 2019 and 2020, respectively. The simulation accuracy of maize LAI, plant height, and biomass during calibration and validation was shown in S4 Fig. Analysis indicated that the SWAP model demonstrated high accuracy in simulating various indicators during the calibration and validation processes. The simulated $R^2$ for various growth indicators ranged from 0.94 to 0.98, while the NRMSE ranged from 8.01% to 14.76%. The RMSE for LAI simulation was 0.44 $m^2/m^2$ and 0.53 $m^2/m^2$, for plant height simulation it was 12.43 cm and 15.81 cm, and for biomass simulation it was 888.02 kg/ha and 1437.76 kg/ha, respectively. The consistency and accuracy of the simulation were performed well, and the simulation results reached a very high level of accuracy.

The simulation accuracy of maize LAI, plant height, and biomass for each treatment during calibration and validation was shown in S3 Table. According to the analysis, the $R^2$ of LAI simulation for each treatment ranged from 0.92 to 0.97, the RMSE ranged from 0.35 $m^2/m^2$ to 0.65 $m^2/m^2$, and the NRMSE ranged from 9.51% to 14.45%. The $R^2$ of plant height simulation for each treatment ranged from 0.93 to 0.99, the RMSE ranged from 9.63 cm to 27.01 cm, and the NRMSE ranged from 5.72% to 18.15%. The $R^2$ of biomass simulation for each treatment ranged from 0.93 to 0.99, the RMSE ranged from 636.65 kg/ha to 1986.33 kg/ha, and the NRMSE ranged from 9.51% to 14.45%. It could be seen that the accuracy of simulating maize LAI, plant height, and biomass for each treatment was high to extremely high during model calibration and validation. The established SWAP model parameter set was reliable to some degree for simulating various growth indicators of maize.

### 2.2.2 Simulation of yield indicators.
The simulation results of maize yield during the calibration and validation of the SWAP model were shown in S5 Fig. During the calibration, the $R^2$ for yield simulation was 0.69, the RMSE was 1272.02 kg/ha, and the NRMSE was 15.59%. The absolute error of yield simulation for each treatment ranged from 443.80 kg/ha to 2160.48 kg/ha, and the relative error ranged from 8.71% to 22.23%. During the validation, the $R^2$ was 0.53, the RMSE was 18078.9 kg/ha, and the NRMSE was 18.37%. The absolute errors ranged from 2374.1 kg/ha to 27021.8 kg/ha, and the relative errors ranged from 4.38% to 29.21%. The simulation results all achieved a relatively high level of accuracy. However, the RMSE was relatively large, indicating that there were outliers in the simulation results of individual treatments. In the calibration, the simulation errors for T6 and T7 were large, while the simulation accuracy was the highest from T1 to T5. In the validation, the simulation errors from T4 to T6 were relatively large, but the simulation accuracy from T1 to T3 and from T7 to T9 was high. This might have been because the rainfall difference over the two years led to varying degrees of water stress on maize under the same irrigation treatment. Due to the model's failure to account for the impacts of various stress factors, such as soil salinity, spatial variability of soil properties, weeds, pests, and diseases, on yield, as well as issues related to parameter calibration, including heteroscedasticity and overfitting, along with the model's assumptions regarding the parameterization of maize growth processes, the model could not achieve a high degree of simulation accuracy for the maize yields of all treatments. Consequently, this led to significant errors in some simulated yield values and a reduction in the overall simulation consistency for the yields of all treatments. To verify the model's reliability in yield simulation, it was necessary to conduct multi-point and multi-year field experiments to further assess the model's stability.

### 2.2.3 Independent validation and confidence interval analysis.

(1) Overview of the experimental area and experimental design

The simulation accuracy of the SWAP model for maize yield was further validated using the irrigation experiment conducted in Pingluo County, Ningxia from 2022 to 2023. The experimental area was located in Baofeng Town (E106 ° 43 ′, N39 ° 02 ′), with a temperate continental climate. The average annual rainfall was 177 mm, the average annual evaporation was 1755 mm, the average annual sunshine hours were 3008 h, and the frost-free period was 171 d. The soil in the experimental area was sandy loam, having an average field water holding capacity of 27% and an average bulk density of 1.48 g/cm$^3$. The available nitrogen, phosphorus, and potassium contents, as well as the organic matter content in the cultivated layer of the experimental site, are 74.36 mg/kg, 13.21 mg/kg, 114.28 mg/kg, and 1.9%, respectively. The experiment was a single-factor experiment on irrigation amount, using drip irrigation as the irrigation method. Five gradient irrigation amounts were set up with the conventional irrigation amount of local farmers as the control, and the irrigation quotas were 2400 m$^3$/ha (S1), 2700 m$^3$/ha (S2), 3000 m$^3$/ha (S3), 3300 m$^3$/ha (S4), 3600 m$^3$/ha (S5). The division of experimental plots, fertilization, indicator determination methods, and other field management measures were the same as before. The irrigation amount for each growth period is shown in S4 Table.

(2) Validation results and confidence interval analysis.

The simulation and confidence-interval analysis results of the SWAP model for maize experimental yield were presented in S6 Fig. The analysis revealed that the SWAP model had a simulated $R^2$ of 0.71, an RMSE of 921 kg/ha, and an NRMSE of 11.62% for maize yield in Pingluo County, indicating good simulation accuracy. Through the confidence-interval analysis of the measured and simulated values of all yield in Yongning County and Pingluo County, it was observed that many yield values fell within the 95% confidence band, and all yield values were included in the 95% prediction band, with an $R^2$ of 0.70, an RMSE of 1227.59 kg/ha, and an NRMSE of 14.58%. Overall, the simulation accuracy of the SWAP model for maize yield could meet general needs and had certain applicability. Therefore, the established maize SWAP model parameter set had a certain degree of scientificity and reliability.

## 2.3 Diagnosis of water stress

### 2.3.1 Model calibration process.
The calculation results of water stress coefficients (Ws) for each treatment during the model verification were shown in S7 Fig. There were significant differences in the water stress experienced by maize throughout its entire growth period under different irrigation treatments. However, the overall manifestation was that in the early stage, the water stress was relatively mild and the duration was short. It gradually increased after 80 days of emergence and reached the most severe level at 100 days (the average Ws of each treatment ranged from 0.07 to 0.60), and then the water stress gradually decreased. This stage was from the tasseling stage to the milk ripening stage of maize, during which the water consumption was high. Insufficient irrigation could lead to severe water stress. Maize experienced varying degrees of water stress in each treatment from 20 to 70 days after emergence. The average Ws of each treatment decreased from values within the range of 0.99 to 1.0 to values within the range of 0.81 to 0.99 after irrigation on days 38, 48, and 56 after emergence, indicating that water stress was caused by excessive irrigation. On the 31st day after emergence, irrigation alleviated water stress in treatments T1 to T6, with the average Ws increasing from values within the range of 0.83 to 0.95 before irrigation to values within the range of 0.99 to 1.0. However, in treatments T7 to T9, it exacerbated water stress, resulting in a decrease in Ws. On the 69th day after emergence, irrigation alleviated water stress in treatments T1 to T4, with the average Ws increasing from values within the range of 0.90 to 0.96 before irrigation to values within the range of 0.99 to 1.0, but exacerbated water stress in treatments T5 to T9. It could be

seen that the maize irrigation scheduling set during the model verification process was unreasonable and needed to be optimized based on water stress conditions.

**2.3.2 Model validation process.** The calculation results of water stress coefficients (Ws) for each treatment during model validation were presented in S7 Fig. There were varying degrees of water stress in the early stages of maize growth for treatments T1 to T3. A severe and long-lasting water stress occurred from 70 to 120 days after emergence, with the most intense water stress occurring at 92 days. The Ws values of each sample ranged from 0.18 to 0.47. After irrigation, water stress was alleviated. However, due to the high water consumption from the tasseling stage to the milk ripening stage of maize, severe water stress persisted after irrigation. During the entire growth period of maize, treatments T4 to T9 experienced different degrees of water stress on 10 occasions, and the duration of water stress was closely related to the amount of irrigation. Among them, treatments T4 to T6 had a longer duration of water stress from the tasseling stage to the milk ripening stage of maize, while the duration of water stress episodes in other treatments was relatively shorter. Water stress first emerged on the 32nd day after emergence, and irrigation mitigated the water stress conditions for each treatment. For the 2nd to 6th, as well as the 8th and 10th water stress events, the average Ws before irrigation ranged from 0.98 to 1.0, and it ranged from 0.76 to 0.98 after irrigation, indicating that these water stress events resulted from excessive irrigation. For the 7th water stress event, the pre-irrigation Ws values of treatments T4–T6 ranged from 0.70 to 0.94, and these treatments were relieved to a state without water stress after irrigation. However, the average Ws values of treatments T7–T9 were all 0.99 before irrigation, which was attributed to excessive irrigation. During the 9th water stress event, the average Ws of each treatment ranged from 0.87 to 0.90 before irrigation and increased to 0.99 to 1.0 after irrigation. During model validation, the irrigation scheduling also exhibited some unreasonable issues. Therefore, it was necessary to further optimize the irrigation scheduling based on the water consumption pattern of maize.

## 2.4 Optimization of irrigation scheduling

The irrigation amount, yield, and irrigation water utilization efficiency (IWUE) of the optimized and original irrigation schemes for maize at each stage were shown in S8 and S9 Figs, respectively. The simulated irrigation amounts obtained from the three optimized irrigation schemes in 2019 and 2020 ranged from 345.12 $m^3$/ha to 462.55 $m^3$/ha. These amounts were at the T4-T6 level of the original irrigation scheme and were reduced by 20.93%− 41.01% compared to those of the T9. After optimizing the allocation of water resources, the simulated maize yields obtained from the three irrigation schemes over the two -year period ranged from 10069.8 kg/ha to 14734.8 kg/ha, the differences among all treatments reached a highly significant level (P<0.01), indicating that there were statistically significant inter-group differences. Among them, the O1 irrigation scheme yielded the highest output, with 14734.8 kg/ha in 2019 and 14596.65 kg/ha in 2020, which were 10.38% and 9.67% higher than those of the T9, respectively. The O2 treatment resulted in lower yields compared to the T6-T9 treatments, while the O3 treatment produced yields that were 2.57% and 1.71% higher than those of the T9 in 2019 and 2020, respectively. The three optimized irrigation schemes all improved the IWUE of maize. The simulated IWUE of the three irrigation schemes over the two-year period ranged from 2.82 kg/$m^3$ to 3.37 kg/$m^3$, except for a few treatments (for example, T6 and T7), the differences between most of the other treatments reached a highly significant level (P<0.01). The O3 scheme had the highest IWUE, which increased by 37.39% and 22.68% respectively compared to that of the T7 (the original maximum value). This was followed by the O1 scheme, which showed an increase of 32.73% and 20.22% in IWUE respectively compared to that of the T7. Analysis indicated that optimizing irrigation could not only reduce the irrigation water amount but also increase maize yield and IWUE. In arid and semi-arid areas, adopting the O3 irrigation scheme could improve water resource utilization efficiency without reducing yield.

## 3. Discussion

### 3.1 Practical evaluation of SWAP model

The SWAP model was a water-driven mechanistic model that had been widely used for simulating the growth of maize in arid regions. Jiang et al. (2016) employed the SWAP model to simulate maize growth in the arid inland areas of Northwest

China. The results demonstrated that the model validation process had yielded good simulation effects on maize relative yield, soil moisture, and soil salinity, with $R^2$ values exceeding 0.8 [26]. Pan et al. (2020) found that the SWAP model also had high accuracy in simulating the growth process of maize in saline-alkali soil. During the model calibration process, the MRE values for simulating soil moisture content, soil salinity, and yield ranged from 15% to 20% [27]. Yuan et al. (2019) also found that the calibrated SWAP model could effectively simulate the dynamic changes in soil moisture and salinity, as well as maize yield, in the saline – alkali areas of northwest arid regions. The MRE simulated for each indicator was less than 20%, achieving a high level of accuracy [28]. Meanwhile, the SWAP model could also simulate the growth process of maize under complex conditions, such as intercropping patterns and areas with shallow groundwater levels [29,30]. In addition, Hassanli et al. (2019) compared the simulation effects of multiple models on maize yield and found that the SWAP model had a certain accuracy advantage in saline-alkali areas [31]. In summary, the SWAP model, after calibration, could effectively simulate indicators such as maize growth, yield, and soil moisture content. It could be used for numerical simulations in fields such as optimizing maize irrigation systems, adjusting cultivation management, and breeding.

### 3.2 Optimization of irrigation scheduling

Crop growth models have been widely used in optimizing maize irrigation schedules due to their strong adaptability, high efficiency, and low cost. Li et al. (2020) developed an irrigation simulation optimization model to optimize the irrigation scheduling with the goal of achieving maximum yield and minimum ET for maize. It was proposed that the irrigation amount for maize in different rainfall year types in the arid areas of the Heihe River Basin should be 147 mm-341 mm [32]. Chen et al. (2020) combined the DSSAT model with an optimization algorithm for irrigation schedules during the growth period based on daily predicted maize yield trends. The difference between the predicted yield and potential yield was used to determine the irrigation amount. In the experimental year, the irrigation amount was reduced by 55 mm-197 mm, achieving the dual goals of water conservation and yield increase [33]. Wang et al. (2020) optimized the sowing time and irrigation scheduling of maize using the APSIM model, taking into account yield, irrigation efficiency, and mechanical harvesting requirements. The optimal irrigation amount was determined as 240 mm (divided into three irrigation sessions) [34]. These research results differed significantly from the irrigation amount proposed in this article, mainly because the optimization of maize irrigation scheduling in different regions was often constrained by local water resource conditions, field management practices, and climatic factors. Consequently, the simulation process differed in terms of constraint settings, cost functions, and optimization objectives. For instance, in water-scarce areas, irrigation quotas, water-use efficiency, or actual evapotranspiration ($ET_a$) were usually treated as the primary constraints, even if this entailed a certain degree of yield loss. In areas with convenient field management, both the number of irrigation events and the irrigation amount were typically set as variables; suitable irrigation scheduling was identified through extensive simulations in a two-dimensional space. In regions with mechanized harvesting, parameters related to the maize growth period were adjusted to maximize rainfall utilization and reduce irrigation water demand, thereby developing water management strategies that balanced rainfall and artificial irrigation. Additionally, the maize irrigation scheduling proposed in these studies often included additional conditions such as adjustments to sowing dates and allowances for yield loss. In the Yellow River Irrigation Area of Ningxia, Wang et al. (2024) found through field experiments that the maximum maize yield had been achieved when the irrigation amount was 539 mm [35]. Zhang et al. (2023) found through long-term meteorological data analysis that the maize water consumption in this area had been 452.9 mm [36]. These conclusions were close to the findings of this study. Because the irrigation amount proposed in study was based on the assumption that maize did not experience water stress or only experienced mild water stress, the aim was to minimize the loss of maize yield due to water stress. From this, it could be seen that the irrigation scheduling for maize had strong regional and purposive characteristics. Reasonable irrigation scheduling was developed based on regional characteristics and management requirements to achieve multiple goals of improving maize yield and water use efficiency.

### 3.3 Limitations of this study

The irrigation scheduling optimization method proposed in this study lacked practical verification and had not been applied in irrigation decisions for maize production. The irrigation decision results were only based on simulated scenarios from past experimental treatments, and the actual results of yield and growth processes had not been fully validated. In addition, the reliability of the decision needed to be further verified due to errors caused by simplification assumptions, parameter value errors, and other reasons.

## 4. Conclusion

This study utilized the SWAP-IES assimilation simulation system to simulate the changes in the water stress coefficient during the growth period of maize under different irrigation treatments in the arid northwest region of China, diagnose the water stress status of maize, determine the period with the highest water consumption of maize, and optimize the irrigation scheduling based on the diagnostic results. The simulation results showed that the optimized irrigation scheduling could reduce the amount of irrigation while maintaining a high yield of maize, and greatly improve the efficiency of irrigation water production. This study provided an intelligent decision-making method for maize irrigation, which could greatly reduce the amount of irrigation water for maize, improve the utilization efficiency of irrigation water, and provide support for efficient water use for maize in arid areas.

### Supporting information

**S1 Fig. The meteorological parameters of the experimental area.**
(PDF)

**S2 Fig. Flow chart of SWAP-IES assimilation system.**
(PDF)

**S3 Fig. The sensitivity of maize LAI, plant height, and yield to the main parameters of the SWAP model.**
(PDF)

**S4 Fig. The simulation accuracy of maize LAI, plant height, and biomass during calibration and validation.**
(PDF)

**S5 Fig. The simulation accuracy of maize yield during calibration and validation.**
(PDF)

**S6 Fig. Simulation and confidence interval analysis of SWAP model on maize experimental yield.**
(PDF)

**S7 Fig. The water stress coefficients (Ws) of each treatment during the model calibration and verification, the values in the figure represent the mean values simulated for each ensemble.**
(PDF)

**S8 Fig. Optimized irrigation amount for each growth stage of maize and simulated yield, the analysis of variance in Figure 8a and b shows the results by year, upper case letters indicated 1% significance level, lower case letters indicated 5% significance level.**
(PDF)

**S9 Fig. The irrigation water utilization efficiency (IWUE) of the optimized irrigation scheduling and the original irrigation scheduling.** Upper case letters indicated 1% significance level, lower case letters indicated 5% significance level, and the blue fonts T1 - T9 and O1 - O3 below the data point were the number of processing or simulation scenarios.
(PDF)

**S1 Table. Irrigation amount for each treatment at different growth stages of maize in Yongning (m$^3$/ha).**
(PDF)

**S2 Table. The range of Ws values for maize at different growth stages during the optimization process of irrigation scheduling.**
(PDF)

**S3 Table. The simulation accuracy of maize LAI, plant height, and biomass for each treatments during calibration and validation.**
(PDF)

**S4 Table. Irrigation amount for each treatment at different growth stages of maize in Pingluo (m$^3$/ha).**
(PDF)

**S1 Appendix. Main basic crop parameter values and sources of maize in SWAP model.**
(PDF)

## Author contributions

**Conceptualization:** Jianxin Jin.

**Data curation:** Jianxin Jin.

**Formal analysis:** Jianxin Jin.

**Funding acquisition:** Jianxin Jin.

**Investigation:** Jianxin Jin.

**Methodology:** Jinyin Lei.

**Project administration:** Jinyin Lei.

**Resources:** Jinyin Lei.

**Software:** Jinyin Lei.

**Supervision:** Jinyin Lei.

**Validation:** Lina Zhou.

**Visualization:** Lina Zhou.

**Writing – original draft:** Lina Zhou.

**Writing – review & editing:** Lina Zhou.

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
