## [Decision Letter · Decision Letter 0]

1 Oct 2025

Dear Dr. Jin,

Thank you for submitting your manuscript to PLOS ONE. After careful consideration, we feel that it has merit but does not fully meet PLOS ONE’s publication criteria as it currently stands. Therefore, we invite you to submit a revised version of the manuscript that addresses the points raised during the review process.

**ACADEMIC EDITOR:** 

**Please address all comments raised by both reviewers below.**

We look forward to receiving your revised manuscript.

Kind regards,

Ahmed Kheir, PhD

Academic Editor

PLOS ONE

Journal Requirements:

[Ningxia Agricultural Science and Technology Independent Innovation Special Science and Technology Innovation Guidance Project (NKYJ-25-03)].

4. In the online submission form, you indicated that [Data cannot be shared publicly because of support project is currently being implemented. Data are available from the Ningxia Academy of Agricultural Sciences Institutional Data Access for researchers who meet the criteria for access to confidential data].

Reviewers' comments:

Reviewer's Responses to Questions

**Comments to the Author**

1. Is the manuscript technically sound, and do the data support the conclusions?

Reviewer #1: Yes

Reviewer #2: Yes

2. Has the statistical analysis been performed appropriately and rigorously?

Reviewer #1: Yes

Reviewer #2: Yes

3. Have the authors made all data underlying the findings in their manuscript fully available?

Reviewer #1: Yes

Reviewer #2: No

4. Is the manuscript presented in an intelligible fashion and written in standard English?

Reviewer #1: No

Reviewer #2: Yes

Reviewer #1: 1.The study applies the SWAP-IES system for irrigation optimization, but it is not entirely clear what is novel beyond integrating previously established methods. Similar work exists on combining crop models with data assimilation to optimize irrigation (eg, APSIM, DSSAT with Ensemble Kalman Filter/IES approaches). The authors should explicitly clarify:

- How their approach is innovative compared to previous SWAP-based irrigation optimization studies. Please cite and use the following papers (https://doi.org/10.1016/j.agwat.2025.109668;
https://doi.org/10.3390/w14223647;
https://doi.org/10.1016/j.agwat.2021.107122)

- Whether the Ws-based irrigation thresholds (O1–O3) are new or adapted from literature, and if adapted, how they are improved here.

2. Calibration/validation results for LAI, biomass, and plant height are strong, but yield simulation accuracy is only moderate (R²=0.53–0.69, with large RMSE). This weak yield performance should be addressed. Discuss potential causes (parameterization, model structural limits, lack of physiological constraints, or field variability). Consider using multi-year or cross-site calibration to improve robustness. Quantify uncertainty more rigorously (eg, confidence intervals, bootstrapping).

3. The choice of Ws thresholds (0.90–0.95) seems arbitrary. There is no clear agronomic justification or sensitivity analysis showing why these values were selected above other possible thresholds. Suggest performing a sensitivity analysis to demonstrate the effect of varying Ws thresholds on yield, IWUE, and total irrigation.

4. The authors acknowledge that the optimized schedules have not been field-tested. This is a significant limitation.

5. The discussion includes some literature comparison, but the differences in irrigation amounts between this study and others (eg, APSIM/DSSAT outputs) need deeper analysis. Are the differences due to model choice, parameterization, soil/meteorological conditions, or the inclusion of salinity leaching requirements in this region?

6. The Yellow River Irrigation Area has salinity challenges, yet the model does not appear to incorporate salinity effects in simulating yield or water stress.

7. No statistical significance testing (eg, ANOVA) is provided for differences in yield or IWUE among treatments, which limits confidence in the claims of superiority for O1–O3

8. Figures 6 and 7 (Ws patterns) are overly complex, and the color scheme and legend are difficult to interpret.

9. Figures 8 and 9 should include statistical error bars and significance groupings.

Reviewer #2: I want to thank the authors' team for producing this valuable article, it is very well written, but I have some comments to furtherly improve it.

In the title, you need to clarify the geographical location of the study at least the country name.

In line no. 9, you need to correct the word "was" by the word "is".

In line no. 35, you need to substitute the word "effects" with the word "impacts" as it has more relation with climate conditions.

In line no. 36, please remove the repeated "had" and complete the sentence by mentioning the good effect of irrigation on what?

In line no. 116, please add the word "sowing" after "The" at the beginning of the line, then correct the phrase "maize sowing plants" to be "maize sown plants".

In line no. 122, please put "in" instead of "with".

In line no. 131, you need to correct the sentence to be "Additional testing has been done after rainfall, which is .....". Please mention the testing type or method.

In line no. 132, please clarify at which age the plant height has been measured, and how frequent measurements were done.

In line no. 138, how many measurements have been taken?

In line no. 139, please put after":" the beginning of the sentence which is "It was...".

In line no. 142, put the "physiological maturity" instead of "maturity".

In line no. 143, put "cm" after the number "10".

In line no. 144, correct the sentence to be "and plants and ears were counted".

In line no. 148, please correct the sentence to be "...and it was converted into the yield (kg) per hectare.

In line no. 151, please remove "The" at the beginning and put "is" instead of "was".

In line no. 162, make the "a" for "accumulated" in lower case.

In line no. 163, after ":" put "They were"

In line no. 427, after 20.22% write "in IWUE".

In line no. 491, add the location at the end of the sentence.

.

Reviewer #1: No

Reviewer #2: No

---

## [Author Response · Author response to Decision Letter 1]

10 Dec 2025

Thank you very much for the hard work of the reviewers and editors. We have carefully studied the review comments, made significant revisions to the manuscript, and provided one-on-one responses to the review comments. Please refer to the attachment for details.

---

## [Decision Letter · Decision Letter 1]

27 Feb 2026

Optimization of Irrigation Scheduling for Maize in Arid Regions Northwest China Based on Water Stress Diagnosis in Models

PONE-D-25-40460R1

Dear Dr. Jin,

We’re pleased to inform you that your manuscript has been judged scientifically suitable for publication and will be formally accepted for publication once it meets all outstanding technical requirements.

Kind regards,

Miquel Vall-llosera Camps

Senior Staff Editor

PLOS One

Reviewers' comments:

Reviewer's Responses to Questions

**Comments to the Author**

Reviewer #1: All comments have been addressed

2. Is the manuscript technically sound, and do the data support the conclusions?

Reviewer #1: Yes

3. Has the statistical analysis been performed appropriately and rigorously?

Reviewer #1: Yes

4. Have the authors made all data underlying the findings in their manuscript fully available?

Reviewer #1: Yes

5. Is the manuscript presented in an intelligible fashion and written in standard English?

Reviewer #1: Yes

Reviewer #1: The authors have been addressed all comments and the manuscript could be accepted for the publication.

.

Reviewer #1: **Yes:** Marwa G. M. AliMarwa G. M. AliMarwa G. M. AliMarwa G. M. Ali

---

## [Editor Report · Acceptance letter]

PONE-D-25-40460R1

PLOS One

Dear Dr. Jin,

I'm pleased to inform you that your manuscript has been deemed suitable for publication in PLOS One. Congratulations! Your manuscript is now being handed over to our production team.

Kind regards,

on behalf of

Dr. Miquel Vall-llosera Camps

Staff Editor

PLOS One